# Predicting thickness perception of liquid food products from their non-Newtonian rheology

Antoine Deblais [1,6]✉, Elyn den Hollander[1], Claire Boucon[1], Annelies E. Blok [2], Bastiaan Veltkamp[3], Panayiotis Voudouris[1], Peter Versluis[1], Hyun-Jung Kim[1], Michel Mellema[1], Markus Stieger[2,4], Daniel Bonn[3] & Krassimir P. Velikov[1,3,5]✉

The "mouthfeel" of food products is a key factor in our perception of food quality and in our appreciation of food products. Extensive research has been performed on what determines mouthfeel, and how it can be linked to laboratory measurements and eventually predicted. This was mainly done on the basis of simple models that do not accurately take the rheology of the food products into account. Here, we show that the subjectively perceived "thickness" of liquid foods, or the force needed to make the sample flow or deform in the mouth, can be directly related to their non-Newtonian rheology. Measuring the shear-thinning rheology and modeling the squeeze flow between the tongue and the palate in the oral cavity allows to predict how a panel perceives soup "thickness". This is done for various liquid bouillons with viscosities ranging from that of water to low-viscous soups and for high-viscous xanthan gum solutions. Our findings show that our tongues, just like our eyes and ears, are logarithmic measuring instruments in agreement with the Weber-Fechner law that predicts a logarithmic relation between stimulus amplitude and perceived strength. Our results pave the way for more accurate prediction of mouthfeel characteristics of liquid food products.

[1] Unilever Innovation Centre Wageningen, Wageningen, The Netherlands. [2] Food Quality and Design, Wageningen University, Wageningen, The Netherlands. [3] Van der Waals-Zeeman Institute, IoP, University of Amsterdam, Amsterdam, Netherlands. [4] Division of Human Nutrition and Health, Wageningen University, Wageningen, The Netherlands. [5] Soft Condensed Matter, Debye Institute for NanoMaterials Science, Utrecht University, Utrecht, The Netherlands. [6] Present address: Van der Waals-Zeeman Institute, IoP, University of Amsterdam, Amsterdam, Netherlands. ✉email: A.Deblais@uva.nl; Krassimir.Velikov@unilever.com

Mouthfeel of food products is a sensory determinant of product liking and repeat purchase. Prediction of mouthfeel sensory attributes (e.g. thickness, creaminess) of food products is of paramount importance for the food industry but has proven to be difficult. It is usually attempted to link rheological parameters (e.g. shear viscosity, storage, and loss modulus[1–3]) and tribological properties[4–6] with mouthfeel sensory perceptions by a trained test panel. Simple models, however, do not accurately describe the complex processes that liquid foods undergo in the mouth[7–12], making it difficult to provide a deeper understanding of the relationship between the mouthfeel of liquid foods and its rheological properties. Importantly, the rheology of food products is rarely that of a Newtonian fluid with a viscosity independent of shear rate. This complicates the modeling of the complex (e.g. elongational) flow[13] and lubrication[14,15] in the oral cavity. Other complicating factors are interactions of the foods with saliva[16,17], wetting of, and food deposition on the tongue and in the oral cavity[10]. The presence of particles also contributes[18,19] to mouthfeel, with particle concentration, size, shape, and hardness[20] affecting mouthfeel. When it comes to the descriptive sensory evaluation of a food texture, subjects can also have difficulties to properly assess a specific attribute because of the difficulty to disentangle a texture attribute from another[21]. For instance, "stickiness" refers to the sensation of food sticking to the palate and tongue during oral processing, but it is possible that subjects also assess this as "cohesiveness", which is thought to be critical for the initiation of swallowing[22].

These complexities have prevented a complete understanding of even the simplest and yet one of the most important sensory attributes of liquid foods: the perceived "thickness". The mouthfeel "thickness" is mostly linked to the viscosity of the food product, and sensory panels are relatively good in assessing the thickness of different liquid food substances[23–25]. It is generally assumed that to sense the thickness of low viscosity solutions while eating, we apply a minimum stress and increase the rate of deformation; for highly viscous foods, the deformation rate is maintained while the stress is increased[26,27]. This means that for low viscosity products, smaller absolute differences in viscosity are more easily detectable than for higher viscous products[28]. This observation is part of a larger debate in the psychosensory field on how the "strength" of a sensation is perceived by humans. The Weber-Fechner law states that the perceived sensory intensity (i.e. thickness) of a stimulus is proportional to the logarithm of the physical stimulus intensity (i.e. viscosity, $S = k \cdot \log(I)$)[28,29]. This relation is found in numerous studies for various senses[30–32], including thickness of very viscous products[28,33]. An alternative scaling has been proposed, known as Stevens' law[34,35] that describes a power law relationship between the perceived intensity and the physical stimuli ($S = k \cdot I^n$). While there is still a debate over Weber-Fechner vs Stevens' laws, more

recent work using a mathematical approach such as a Bayesian framework supports a logarithmic scaling of the response to a stimulus[36–38] for auditory and number perception. However, when it comes to the question of mouthfeel, little is known about the psychophysical response of our tongue to the process of swallowing food. This is mainly because for liquid food products with a low non-Newtonian viscosity that depends on the stress and deformation rate, it is not clear under what conditions the food viscosity needs to be evaluated in order to relate it to the mouthfeel. For a long time it has been assumed that there is a single relevant shear rate in the oral cavity and many attempts have been made to correlate the perceived product thickness to the viscosity at this shear rate, also including rudimentary notions of shear thinning[26,27,39,40].

Here we go beyond these simple models and develop a conceptually physical model describing the fluid mechanics of liquid foods in the oral cavity in order to establish relationships between rheological properties of shear-thinning, low viscous liquid foods and the mouthfeel perception of their thickness. To do so, we develop a physical model that captures the fluid mechanics, spreading, and rheology, as well as the biophysical aspects of eating and sensing. We test this model by investigating the rheology and sensory assessment by a test panel of 14 liquid bouillons varying in viscosity by three orders of magnitude, from 1 mPa s to ~1 Pa s (see Fig. 1). We extend our findings by using recent sensory and rheology data of more viscous xanthan gum solutions[25], Table 1 and Methods Section.

## Results

**Rheology of the liquid bouillons.** First, we examine the rheology of the liquid bouillons. As test liquids, we use custom made (set 1) and as-bought commercial bouillon soups (set 2) with varying viscosity. Salt (NaCl) combined with two different polymers (xanthan gum and starch) give the samples and the soups their consistency; the salt and polymer concentrations of the 14 bouillons and liquid samples investigated are given in Table 1. Figure 1B shows examples of the flow curves of our bouillon samples. We find that the bouillons have a viscosity that decreases sharply with increasing shear rate, and can be well described by the following relationship between shear stress $\sigma$ and shear rate $\dot{\gamma}$:

$$\sigma = \kappa \dot{\gamma}^n, \tag{1}$$

with $\kappa$ the consistency parameter and $n$ the power law index; as listed in Table 1, both parameters are found to depend on concentration, and structure of the liquid. This type of fluid is also referred to as power law fluid[41]. Some of the bouillon samples exhibit very weak viscoelasticity due to the presence of xanthan gum. For what we will consider in the remainder, elastic

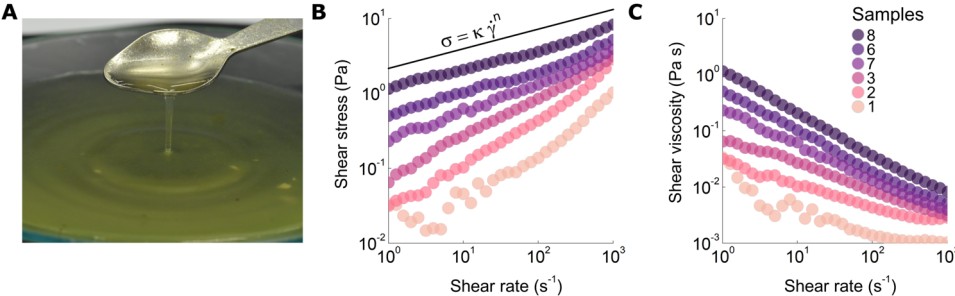

**Fig. 1 Rheology of liquid bouillons. A** Picture of one of the bouillons investigated. **B** Typical flow curves of different samples with the numbering referring to Table 1. They are well described by a simple power law $\sigma = \kappa \dot{\gamma}^n$ (solid line) with consistency parameter $\kappa$ and power law index $n$. Table 1 lists $\kappa$ and $n$ values for all samples studied. Color codes indicate different samples. For readability, not all samples are shown. **C** Corresponding shear viscosities as a function of shear rate for the same bouillon samples as shown in **B**.

**Table 1 Properties and measurement/test outcomes of all samples investigated.**

**Set 1 - Bouillon Custom-made**

| Sample # | Xanthan gum [wt%] | Starch [wt%] | NaCl [wt%] | Thickness score (sensory panel, [ · ]) | $\kappa$ [Pa s$^n$] | $n$ [ · ] |
|---|---|---|---|---|---|---|
| 1 | 0.0 | PS–1.5 | 0.74 | 2.09 (0.17) | 0.026 | 0.55 |
| 2 | 0.3 | PS–1.5 | 0.74 | 2.92 (0.25) | 0.025 | 0.60 |
| 3 | 0.6 | CS–3.0 | 0.76 | 4.25 (0.21) | 0.068 | 0.55 |
| 4 | 0.6 | PS–3.0 | 0.65 | 3.81 (0.29) | 0.590 | 0.26 |
| 5 | 0.8 | PS–4.0 | 0.73 | 3.85 (0.26) | 0.330 | 0.31 |
| 6 | 1.2 | PS–6.0 | 0.76 | 4.73 (0.25) | 0.636 | 0.28 |
| 7 | 1.2 | CS–6.0 | 0.65 | 4.63 (0.17) | 0.410 | 0.35 |
| 8 | 1.8 | PS–9.0 | 0.76 | 5.30 (0.24) | 1.000 | 0.27 |
| **Set 2 - Commercial** | | | | | | |
| 9 | NK | NK | NK | 2.89 (0.21) | 0.057 | 0.48 |
| 10 | NK | NK | NK | 4.41 (0.22) | 0.198 | 0.44 |
| 11 | NK | NK | NK | 3.20 (0.25) | 0.130 | 0.43 |
| 12 | NK | NK | NK | 4.69 (0.21) | 0.170 | 0.46 |
| 13 | NK | NK | NK | 3.26 (0.25) | 0.040 | 0.57 |
| 14 | NK | NK | NK | 5.04 (0.17) | 0.270 | 0.42 |
| **Set 3 - Custom-made, extracted from**[25] | | | | | | |
| 15 | 0.04 | 0.0 | 0.0 | 3.3 (1.0) | 0.03 | 0.65 |
| 16 | 0.10 | 0.0 | 0.0 | 3.4 (0.8) | 0.14 | 0.52 |
| 17 | 0.21 | 0.0 | 0.0 | 4.1 (0.8) | 0.35 | 0.40 |
| 18 | 2.00 | 0.0 | 0.0 | 9.4 (0.4) | 7.00 | 0.25 |
| 19 | 3.40 | 0.0 | 0.0 | 11.7 (0.2) | 22.0 | 0.17 |
| 20 | 4.30 | 0.0 | 0.0 | 12.0 (0.2) | 31.0 | 0.17 |

Sample compositions of set 1 are mixed of raw components of xanthan gum (XG), potato starch (PS), corn starch (CS), and sodium chloride (NaCl). The concentration of fat (palm oil stearin) is fixed (0.4 wt%). The compositions for set 2, are very close to the as-bought commercial samples and are undisclosed (NK). Sample compositions in Xanthan Gum for set 3 are extracted from ref. [25] and are used to validate our model to larger values of shear stress. Thickness scores (scale [0–15]) are indicated as mean values (standard error of the mean). Rheology parameters κ and n are obtained by fitting the flow curves of Fig. 1B using Eq. 1. Thickness scores of set 3 taken from[25] have been transposed as described in the Methods section.

contributions are negligible at these concentrations[42] as confirmed by extensional rheology measurements (see Method sections, Supplementary Notes 1, 2 and Supplementary Fig. 1, 2).

The differences between the various liquid bouillons are mainly due to the thickeners used, xanthan gum, potato and corn starches, and their mixtures with xanthan gum; the shear stress and its variation as a function of the shear rate are highly dependent on their concentration.

**Modeling the perceived thickness.** To develop a model that can predict the mouthfeel sensory attribute thickness based on the rheological behavior of the liquid products in the mouth, it is necessary to understand how we "sense" liquids in our mouth. The basic assumption we make is that the thickness of the fluid is determined by holding some of it between the tongue and the palate (Fig. 2); the thickness is perceived by the transmission of stresses on the mechano-receptors of the tongue, the nerve cells sensitive to touch and pressure[43]. In other words, a person examines the liquid bouillon by pressing an amount against the palate with the tongue and moving it back and forth. In this way, the liquid is deformed. The extent to which the liquid product "resists" this deformation, in other words the shear stress, is perceived as the "thickness" of the product.

Based on the pioneering works of DeMartine[39] and Kokini et al.[27,40,44], we model the oral cavity as two parallel flat plates spaced apart at a distance $h$ from each other (Fig. 2D)). Interestingly, this approximation holds because the tongue is much softer ($E_{tongue}$ = 2.5–10 KPa,[45]) than the palate ($E_{palate}$ = 30–50 MPa,[46]). Due to the presence of the liquid in between, this field is called soft lubrication or elasto-hydrodynamic lubrication[47] (EHL) and one can show that the deformability of the tongue makes that we can consider the palate/tongue system as two parallel rigid plates. The resulting

deformation $\delta$ can be calculated from[48,49]:

$$\delta = \left( \frac{9 F_N^2}{16 R_{tongue} E_{tongue}^2} \right)^{1/3} \qquad (2)$$

Here, $R_{tongue}$ is the radius of curvature of the tongue[45], and $F_N$ is the normal force, also known as a lingual force. Substituting all numbers, we obtain deformation $\delta = 7$ mm and a contact area of radius $R = 2$ cm. This means that the deformation is smaller that the characteristic size of the system $R_{tongue} \sim 5$ cm, in line with our assumption of two parallel plates without significant elasto-hydrodynamic effects.

Thus, in our model, the liquid is pressed between the plates with a normal force $F_N$, while the lateral movement of the tongue at constant speed[43], $V$, provides a shear rate $\dot{\gamma} = V/h(t)$, with the gap between the tongue and the palate $h(t)$ changing in time. The earlier models do not properly incorporate the effect of the tongue speed on the evolution of the gap, but also neglect the complex rheological response of their materials (e.g., presence of yield stress). This calls for a reevaluation of the hydrodynamics of the swallowing process.

We model the evolution of the gap $h(t)$ by adding the effect of the tongue velocity, resulting in what we refer to as dynamic squeezing (see Methods section). Note that we do not assume that there is a single characteristic shear rate, but instead consider in the model the full integrated response of all stresses and shear rates generated during consumption of the bouillon.

In our approach, we consider that the perceived thickness of the food product is proportional to the shear stress $\sigma$ at the surface of the tongue[11]: it is this stress that squeezes out the food product from the gap between the tongue and the palate. Based on the properties of the liquid foods that we consider here, the total viscous stress $\sigma$ can be expressed in the rheological

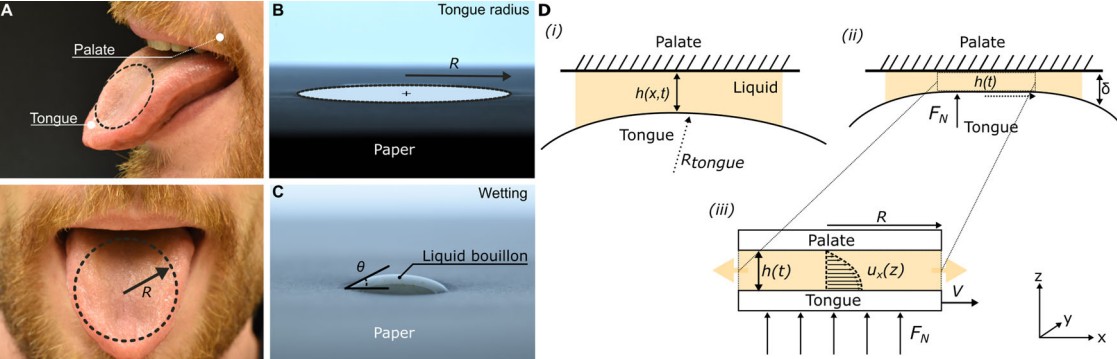

**Fig. 2 Modeling the perceived thickness of liquid bouillons. A** Assessing the thickness of a liquid involves two organs (top): the palate and the tongue. In our approach, we consider that the liquid bouillon covers a circular area of radius $R$ on the tongue (bottom). **B** The maximum radius of coverage $R$ is determined by licking a piece of paper. **C** To determine the wetting properties of the liquid bouillons, small droplets are deposited onto paper and the resulting contact angle $\theta$ is measured. **D** Sketch of the geometry used to model fluid deformation in the mouth (*i*). Because of the softness and deformability of the tongue compared to the palate (*ii*), the two organs can be approximated as two parallel plates separated by a distance $h$ and pressed together with a constant force $F_N$ (*iii*). The bottom plate (tongue) moves at a speed $V$ relative to the top (palate) to deform the trapped liquid, squeezing out the fluid from the (bucal) cavity. $u_x$ stands for the velocity component along $x$, which here varies with height $z$.

parameters $\kappa$ and $n$:

$$\sigma = \kappa \left( \frac{V}{h(t)} \right)^n \quad (3)$$

In the scenario considered here, the tongue is moving at a horizontal speed $V$ while squeezing the liquid with a lingual force $F_N$, giving for the variation of the gap:

$$h(t) = h_0 \left( 1 + \frac{(n+1)F_N h_0^{n+1} V^{1-n} t}{2\pi n \kappa R^4} \right)^{\frac{-1}{n+1}} \quad (4)$$

where $h_0$ is the initial gap between the tongue and the upper part of the oral cavity, $R$ is the radius covered by the liquid product on the tongue (Fig. 2A), $F_N$ is the lingual force, and $t$ is the characteristic time needed for assessment (i.e. the residence time for the liquid product during the sensory test). Different from previously derived models linking rheological properties to thickness perception[27,39,40,44], the complexity of the flow profile is properly taken into account, based on the principle that shearing a shear thinning fluid makes it thinner, which implies that it is squeezed out faster. Note that if one considers the specific case of a Newtonian liquid ($n = 1$), the result becomes independent of the tongue velocity $V$, as expected[50,51]. To solve the equations, we use the fact that the speed of the tongue is higher than the velocity due to the squeezing. We discuss this approximation in more detail in Methods section and in Supplementary Note 3 and Supplementary Fig. 3.

From Eq. (3) we can then evaluate the total stress exerted on the fluid by the tongue as:

$$\sigma = \kappa V^n h_0^{-n} \left( 1 + \frac{(n+1)F_N h_0^{n+1} V^{1-n}}{2\pi n \kappa R^4} t \right)^{\frac{n}{n+1}} \quad (5)$$

Using the total shear stress assures that the effect of all shear rates is considered and, hence, that all dynamic viscosities are considered. This is a major improvement compared to earlier modeling approaches where correlations are made between thickness perception and a value of the viscosity that corresponds to a specific (but arbitrary) shear rate[9].

To use the above model to connect rheology with thickness, we need the pertinent parameters for the mouthfeel evaluation. The panel testing is done with a tablespoon of (semi-)liquid bouillon of volume $V_0 \sim 4$ mL that spreads over the tongue surface and is trapped between the palate and the tongue. The contact radius of the liquid cylinder between tongue and palate is estimated from the

wet area made by licking a piece of paper (where the saliva leaves a stain on the paper that can be measured), and found to be $R = 2.5$ cm (Fig. 2B). A wetting test was also done on the same paper to establish if differences in the wetting properties of the bouillons affect the mouthfeel perception. We find that all 14 bouillons show very similar contact angles, partially wetting[52] the paper surface with a contact angle of $\theta \sim 30°$ (Fig. 2C). This likely results from their similar biopolymer composition and assures that the bouillons all have the same initial condition in the oral cavity (Fig. 2D). From this, we find the initial gap to be $h_0 \sim V_0 / \pi R^2 = 2$ mm, corresponding to the initial height of the liquid bridge. The normal force $F_N$ of the tongue and its velocity $V$ have been measured and shown to be almost constant for a range of low viscosity foods with typical values of 500 mN[43] and 15 cm/s[53], respectively (see Supplementary Note 4 and Supplementary Fig. 4 for the effect of the tongue velocity on the generated stress). As the characteristic time $t$ during which the thickness of the bouillon is assessed in the mouth, we choose 1.2 s for all samples based on biophysical studies conducted on low-viscous products[54,55] similar to the ones studied here. For this time, the approximation made to establish Eq. (4) is fully satisfied (Supplementary Note 3 and Supplementary Fig. 3). As a result of this choice of parameters, the shear stress therefore only depends on the flow parameters $\kappa$ and $n$, which, by means of rheological measurements, have been determined in Fig. 1A.

## Discussion

We are now in a position to evaluate our model that relates the squeeze flow parameters to the perceived thickness of the bouillons and xanthan gum solutions. Figure 3B shows the relation between the "subjective thickness" assessed by the test panel and the total shear stress on the tongue as determined from rheological measurements and the above theory. Fitting our data (liquid bouillons of set 1 & 2) shows that a logarithmic relationship works well (solid black line), but also a power law relation cannot be ruled out (dotted line); it should however be noted that the power-law has an extra fitting parameter. Nonetheless, when plotting the residuals $\chi$ of the two fits (see supplementary note 5) it is hard to decide which gives a better description. In fact, earlier observations by Demartine and Kokini et al. for paste-like foods favored a power law; however, their model clearly underestimates the shear stress, because of the simple approximation flow considered in their work (i.e. static squeezing). In addition in their work the input biophysical parameters were used as adjustable

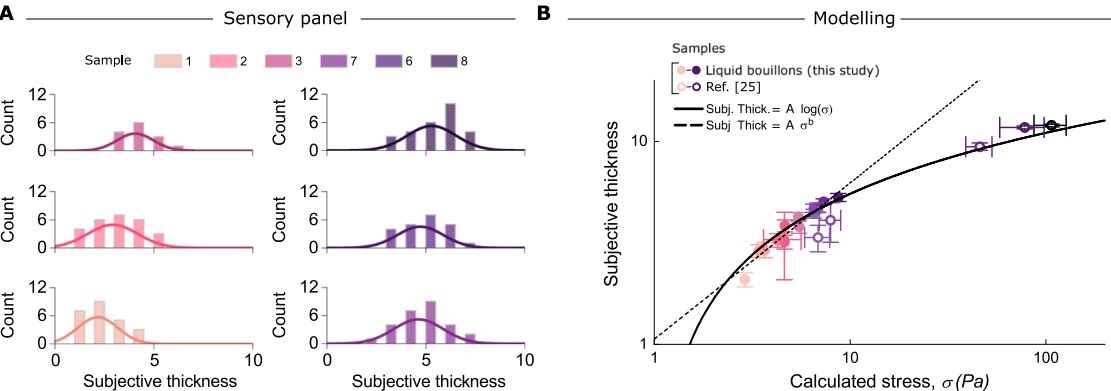

**Fig. 3 Predicting the mouthfeel "thickness" of a thin liquid sample. A** Typical "subjective thickness" distributions obtained from panelists (6 typical samples are shown, same color code as in Fig. 1B, C). Continuous lines are fits to a normal distribution from which mean values and dispersions are obtained, see Table 1. **B** Relation between "subjective thickness" and stress $\sigma$ on the tongue as calculated from our model (Eq. (5)). The black continuous line indicates a logarithmic dependence (Weber-Fechner's law), while the black dotted line shows a power law-dependence (Steven's law). The two fits are obtained by fitting the data of this study (set 1 & 2). Data of set 3 (empty circles) extend the range of stress and confirms the good agreement with the logarithmic dependence found with the previous sets. Error bars are standard error of the mean (SEM).

parameters because they were not known at the time (see Supplementary Fig. 5 for a comparison between the two models).

The solution to the power-law vs log-law discussion is to extend these findings to include also more viscous samples, covering a larger stress range. To do so, we include very recent data by Blok et al.[25] (set 3, Table 1). Those samples were expressly designed to reach large values of shear stress while keeping the elasticity contribution negligible (see the compositions of these solutions in Table 1, and their viscoelasticity in Supplementary Fig. 1). This makes it possible to directly apply our model to these more viscous samples. The key result is that if we include these data in our plot without changing the fitting parameters of the two models for our bouillon samples, we can clearly see that the log dependence on the shear stress (Weber-Fechner law) very well describes all the experimental data (sets 1, 2 & 3), and does much better than the power-law (Stevens), and cover more than a decade of stress.

We conclude that the sensory attribute "thickness", an important element in the mouthfeel of food products, is well predicted with our model which accounts for squeezing flow during liquid food consumption, and which suggests this subjective parameter is proportional to the logarithm of the shear stress on the tongue. This logarithmic relation between perception and stimulus, which has also been observed for other sensory organs such as our eyes and ears, agrees with the Weber-Fechner law. Despite its proven importance and impact on how we perceive things in our daily lives, such psychological law has never been reported before for the oral thickness perception. We are able to conclusively make this connection by considering a more realistic dynamic squeeze flow and by including the biophysical parameters that govern the swallowing process. Interestingly, the values of the calculated stress $\sigma$ reported in this study ($\approx 1-110$ Pa) are in a good agreement with recent studies conducted on simple liquids looking at the deformation of model papillae[11]. Without adjustable parameters, the rheology of a fluid can predict its perceived thickness. Our study shows that the sensory attribute "thickness" is strongly influenced by the shear thinning, as all bouillons are rather strongly shear thinning, with viscosities that vary with shear rate over one to almost two orders of magnitude for a single liquid. The importance of these findings lies in relating the physical properties (i.e. rheology) of liquid food products to their mouthfeel. When coupled with the molecular and structural properties of the ingredients used, such as, for example, thickeners or biopolymers, such an approach may in the future enable us to design food in a way that it achieves the ideal mouthfeel. The results presented here focus on (semi) liquid materials; further experimental and theoretical work is needed towards predicting the mouthfeel of thicker or more complex products such as yield stress fluids with correct hydrodynamics.

## Methods

**Sample preparation.** We prepared sets 1 & 2 (see Table 1) of in total 14 liquid bouillons ranging in viscosity from 1 mPa s to 1 Pa s according to a fixed protocol in a classic saucepan. All samples from the first set were prepared by mixing the raw components: xanthan gum Ginsted® from Dupont, potato/corn starches from Dupont, palm oil stearin fat from Sime d'Arby Unimills B.V. (Zwijndrecht, the Netherlands), and salt from Akzo Nobel (Deventer, the Netherlands) as described in Table 1. Samples have been designed in order to vary significantly by their oral perception. The second set of samples had similar ingredients, but the exact compositions are unknown as the basis of these samples is very close to the commercial samples (Knorr®, Unilever). The different liquid bouillons were cooked in a conventional saucepan (stainless steel, BK®, Excellent series 18 CM). A total of 500 g of pure water MilliQ® was brought to a boil on an electric hob. Once the water was boiling, samples were added and dispersed by using a regular whisker. The solutions were left to simmer (not boil) three more minutes while stirring. The resulting liquid bouillons were filtered using a standard kitchen sieve to prevent aggregates in the bouillons. They were finally placed in a glass container. An oven (Memmert M400) was used to stock the preparations at a constant temperature of 62 °C for further analysis. This temperature is specifically chosen in order to compensate the heat loss during the transfer of the liquid bouillons in the pre-heated soup bowl (see sensory analysis in method section) and thus being served at a temperature very close to the mouth-temperature ~40 °C. For set 3, see[25].

**Sensory analysis.** A strict protocol was followed for the sensory assessment of the samples of sets 1 & 2 by an experienced, highly trained test panel of 11–14 members (experienced in evaluating products, not intentionally all female, average age 54.6 ± 7.4 years old; the panel's gender balance has previously been shown not to influence overall texture perception[56]). Products were offered in a sequential monadic blind test using 3-digit codes. Two measurements were made for each sample, and the product order was randomized per replicate. To ensure minimal product variation (e.g. evaporation resulting in thicker products), panelists received the products in the same order. Breaks were provided in between the samples to allow for palate cleansing. Tap water and unsalted cream crackers were used as palate cleansers. The trained panel received 150 ml of product, served in preheated china soup bowls. Product was prepared directly before serving. The panel testing is done with a tablespoon of volume $V_0 \sim 4$ mL. A modified spectrum method[54] was used with a 16-point category scale (0–15). The panel was trained to score the intensity of all attributes according to salt solution references (see Table 2). The use of absolute scaling enables to compare intensities of attributes to each other. A PROC MIXED analysis (with Respondent and Product as fixed factors, and Respondent × Product as random effect) was performed to see if there were significant differences between the products ($p < 0.05$) on the attributes. Attribute list is reported in Supplementary Table II. Least Square Means were calculated for every attribute for each product. Subsequently, a multiple comparison test was performed to determine which products significantly differed from each other. SAS 9.4 software was used for the statistical analysis. For details about the sensory evaluation of samples of set

**Table 2 Salt scale references for sensory testing with absolute scaling.**

| Score | [NaCl] |
|---|---|
| 3 | 3.5 |
| 6 | 5 |
| 9 | 6.5 |
| 12 | 8 |

3, see[25]. All samples are reported in Table 1. We converted the category scale of this study to ours: ([thickness score XG solution]/10) × 13 + scale reference (Table 2).

### Rheology measurement

*Shear rheology*. The shear rheological properties of the solutions were measured with an Anton Paar series MCR302 rheometer using a cone-plate geometry with 50 mm diameter cone (rough) and a 1° angle. A Peltier cell (P-PTD 200 cell, Anton Paar) allowed the control of the sample temperature during measurements. Rheology measurements were performed at 40 °C, which is the temperature assumed to be the relevant temperature inside the mouth before swallowing, and right after their preparation—as for the sensory analysis—since the samples can change in time due to the presence of starches and xanthan gum (see Supplementary Fig. 6). All measurements were performed three times to ensure reproducibility. To avoid evaporation during rheology measurements, a homemade humidity chamber was fitted around the upper and lower geometries of the rheometer. This allowed to work at a constant humidity (RH = 80%). The injection of a tunable humid air flow in the chamber allowed to suppress evaporation during measurements.

*Extensional rheology*. The extensional properties of the solutions have been measured based on the principle of filament stretching experiment. A rheometer (Anton Paar, MCR 302) was used as the building block of the device. A rheometer geometry plate with a diameter of 5 mm was used, the lower plate having the same diameter. The upper plate can be pulled vertically at a constant velocity until the capillary bridge breaks. A Peltier cell allows us to impose the temperature of the sample during the elongational process at a constant speed. The device is coupled to a high-speed camera (Phantom V701, Vision research) equipped with a microscope lens (Navitar). The profile of the neck diameter is automatically followed in time with a homemade Matlab routine by processing the raw image. More details and results are provided in Supplementary Fig. 1 and Supplementary Note 1.

### Wetting properties

To assure that the investigated liquids are initially trapped between the palate and the tongue, we measured the contact angle of a drop deposited on a piece of paper, with the help of a Drop Shape Analyzer from Krüss. For the measurements, we deposited 1 μL of each liquid samples. We find that the contact radius remains pinned and constant for all samples $\theta \sim 30°$, which is a likely result since they have similar biopolymer composition.

### Dynamic squeezing model

The dynamic squeezing model used to predict the subjective thickness shown in Fig. 3 is derived as follows. For the geometry, we consider an infinite rigid flat plate at the bottom, moving with velocity $V$ in $x$-direction, and a rigid disk of radius $R$ at the top, coming down with velocity $\dot{h}$. As mentioned in the main text, reducing the mouth as two parallel plates is possible because of the deformation of the tongue (Young modulus $E_{tongue} \sim 2.5 - 10$ KPa) when put in contact with the palate $E_{palate} \sim 30 - 50$ MPa. We assume the "thinning effect" causes the flow to be mainly in the $x$-direction, so $u_y = 0$.

The momentum equation for a shear-thinning liquid is found by using the classical lubrication equation[57,58] since the layer thickness $h(t)$ is much smaller than the tongue radius $R$. Here, the $z$-derivatives are dominant, and pressure $P$ is a function of $x$ only:

$$\frac{\partial P}{\partial x} = \frac{\partial \tau_{zr}}{\partial z} = \kappa \frac{\partial}{\partial z}\left(\frac{\partial}{\partial z}u_x\right)^n, \quad (6)$$

with $\kappa$ and $n$ the rheology parameters defined as in Eq. (3). Using the no-slip boundary conditions

$$\begin{cases} u_x(z=0) &= V \\ u_x(z=h) &= 0, \end{cases} \quad (7)$$

we find for the velocity profile:

$$\begin{cases} u_x &= \frac{hn}{n+1}\sqrt[n]{\frac{h}{\kappa}\left|\frac{\partial P}{\partial x}\right|}\left((c_1+\frac{1}{2})^{\frac{n+1}{n}} - (c_1-\frac{z}{h})^{\frac{n+1}{n}}\right), \partial_x p > 0 \\ u_x &= \frac{hn}{n+1}\sqrt[n]{\frac{h}{\kappa}\left|\frac{\partial P}{\partial x}\right|}\left((c_1+\frac{z}{h})^{\frac{n+1}{n}} - (c_1-\frac{1}{2})^{\frac{n+1}{n}}\right), \partial_x p < 0 \end{cases} \quad (8)$$

Here, $c_1$ is a constant determined by the second boundary condition. It is impossible to solve this analytically[59]. To solve the equations, we use the fact that the speed of the tongue is higher than the velocity due to the squeezing, $V >> \frac{hn}{n+1}\sqrt[n]{\frac{h}{\kappa}\frac{\partial P}{\partial x}} \equiv \alpha$. We

discuss this approximation in more detail in Supplementary Note 3 and Supplementary Fig. 3. We thus can use a first-order Taylor expansion of the term within brackets to find (Supplementary Note 6):

$$c_1 \approx \left(\frac{n}{n+1}\right)^n \frac{V^n}{\alpha^n} \quad (9)$$

The flux at any position $x$ is given by:

$$\Phi_1 = \int_{-\frac{h}{2}}^{\frac{h}{2}} u_x \, dz \approx \frac{hV}{2} - \frac{h^{2+n}V^{1-n}}{12\kappa n}\frac{\partial P}{\partial x}, \quad (10)$$

where we use a Taylor expansion and take the leading terms in $V$. This flux should be equal to the flux coming from the movement of the tongue, and the flux due to squeezing:

$$\Phi_2 = \frac{hV}{2} - \dot{h}x \quad (11)$$

Equating these two fluxes yields a differential equation for the pressure. Applying the boundary condition that the additional pressure is equal to zero (ambient pressure) at the boundaries, yields a solution for the pressure:

$$P(x) = \frac{6\kappa n\dot{h}}{h^{n+2}V^{1-n}}(R^2 - y^2 - x^2) \quad (12)$$

The force is then given by the integral of pressure over the surface area:

$$F_N = \frac{3\pi\kappa n\dot{h}R^4}{h^{n+2}V^{1-n}} \quad (13)$$

This is equal to the load on the disk. Solving for the layer thickness $h$ yields the final expression:

$$h(t) = \frac{h_0}{\left(1 + \frac{(n+1)}{n}\frac{F_N h_0^{n+1}V^{1-n}}{3\pi\kappa R^4}t\right)^{\frac{1}{n+1}}} \quad (14)$$

**Reporting summary**. Further information on research design is available in the Nature Research Reporting Summary linked to this article.

## Data availability
The authors declare that the data supporting the findings of this study are available within the paper and its supplementary information files. Interested parties may contact A.Deblais@uva.nl.

## Code availability
Code to generate the shear stress can be provided upon request. Interested parties may contact A.Deblais@uva.nl.

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

## Acknowledgements

We thank Veronica Galindo-Cuspinera and Max Batenburg for useful insights and fruitful discussions. This research was financially supported by Unilever, NanoNextNL, the European Union's Horizon 2020 research and innovation program under Grant agreement no. 798455—PlantEmulGel, by the Netherlands Organisation for Scientific Research (NWO) in the framework of the Innovation Fund for Chemistry, and by the Ministry of Economic Affairs in the framework of the "TKI/PPS-Toeslagregeling".

## Author contributions

A.D., D.B., and K.P.V. designed the research; A.E.B., E.H., C.B., Pan.V., Pet.V., H.-J.K., and M.M. designed and led the sensory measurements and analyzed the resulting data. A.D. and B.V. modeled the dynamics squeeze flow. A.D. performed the rheology measurements; A.D. analyzed the data; A.D., M.S., D.B., and K.P.V. contributed to the final version of the manuscript.

## Competing interests

The authors declare no competing interest.
