## [Peer Review File · Nature Communications]

REVIEWER COMMENTS

Reviewer #1 (Remarks to the Author):

Review report on the manuscript entitled “How ‘thick’ are ‘thin’ liquid food products? Predicting thickness perception from their rheology” by A. Deblais et al.

In their manuscript, Deblais and coauthors address the general problem of how humans perceive with their oral cavity differences in the texture of food products. It is a problem that directly falls within current researches that aim at elucidating how human senses function and what sets their sensitivity. More specifically, Deblais and coauthors investigate the relationship between the perceived thickness of liquid bouillons that have a shear-thinning rheological behavior and the elicited shear stresses at the surface of the tongue. This relationship is obtained by combining the results of psychophysics experiments with a human panel and the predictions for the shear stress at the surface of the tongue. The latter are obtained by modeling the liquid flow in the gap between the tongue and the palate with lubrication theory and by using actual rheological measurements of the liquids that are tested. The model, in particular, aims at reproducing the testing conditions, by incorporating both squeezing and shearing of the liquid. Deblais et al. find that the intensity of the perceived thickness increases with the calculated shear stress on the surface of the tongue and claim that a logarithmic dependence better fits their data, in agreement with the well-known Weber-Fechner’s law already derived for other modality senses.

Overall, the paper is well written, the problem is well motivated and of interest. However, I feel that the results that are presented here do not constitute a significant advance in the field. They lack some experimental proof, the theoretical derivations are rather limited and detailed comparisons to previous models are missing, all reasons combined that cannot justify publication of this work in such a high-end journal as Nature Communications. I also feel that the results presented here would be more appropriate for publication in a more specialized journal.

More specifically, establishing a relationship between the perceived thickness and the shear stresses elicited at the base of the tongue is not new. Actually, it has already been performed by previous authors, in particular Kokini et al., even though I acknowledge that it was done for other types of food products. Kokini et al. measured an analog power law behavior over almost two decades in the shear stress. I am thus unsure of how the present work is very different from that of Kokini et al. The difference I note though, is the logarithmic functional dependence that Deblais et al. claim to have found. This could have been an important outcome of the current work, but given the error bars on their measurements, the data is however equally well fitted with a power law and a logarithm. Deblais et al. are well aware of it but use the rather weak argument that a fit with a single parameter is somehow “the best choice” than a fit with two parameters. They thus conclude that a logarithmic dependence is evidenced. But properly testing such logarithmic dependence would definitely require additional experiments to increase the range of the stress values, in particular at low shear stress values where sensitivity is high as well as at higher shear stress values.

On the theoretical aspects of this work, I am also skeptical about the novelty of the present derivations as they clearly fall within the same theoretical framework that DeMartine and Cussler had developed for the evaluation of liquids between two fingers. Similarly, the authors solve the lubrication theory equation, however with different boundary conditions to take into account the shear motion of the tongue at velocity V . Even if providing a solution to this equation appears rather tricky and calls for some assumptions, I feel that it does not constitute a major improvement.

In addition, the expression the authors find for $h(t)$ (Eq. 13) involves the velocity of the tongue V . How does this expression compare to the result of DeMartine and Cussler? In particular, am I wrong to assume that in the limit $V \rightarrow 0$ (squeeze flow only), one should recover their result?

The current model also allows the authors to provide magnitudes of the elicited stresses at the base of the tongue. How do these compare with actual biological measurements? How could this model be tested directly and how is it better than previous models? These aspects should be discussed.

Finally, this model does not take into account the fact that the tongue itself is much softer than the palate. However, as the gap height tends to zero, one could expect elasto-hydrodynamics effects to come into play. How would these effects change the overall stress response? I feel that these aspects should also be discussed.

Minor comments

1 – The wording ‘thickness’ appears very soon in the main text (starting with the abstract) and it is only properly defined on page 3 of the main manuscript. To ease the reading, this wording should be defined from the start.

2 – On page 6, it looks like the kappa constant is misplaced in Eq. 5. My understanding is that it should be outside the n powered term.

3 – In deriving Eq. 8, assumption is made that the parameter α is much smaller than the velocity V . Looking at Fig. 4 of the Supplemental Information, this assumption appears to be valid at long times only and not for $t > 0.2s$ as the authors write in Supplementary Note 5. The expression the authors find for $h(t)$ (Eq. 13) is thus, I understand, only valid at long times and this should be clearly stated in the main text.

Reviewer #2 (Remarks to the Author):

This paper proposes a novel method for modelling perceived thickness of thickened liquids by suggesting a logarithmic relationship between perceived thickness and stress on the tongue. This

paper introduces some new and interesting research to the field, and it brings up valid criticisms of current methods for correlating instrumental measurements with perceived thickness. This paper would be suitable for publication with minor revisions as suggested below.

In the introduction, it would be good for the authors to discuss the effect of other textural attributes on the perception of thickness, such as stickiness and slipperiness. In addition, the authors should include that besides rheology, there is work being done with tribology that is also trying to correlate instrumental measures with sensory textural attributes.

In the methods, the authors mention that “the panel was trained to score the intensity of all attributes”. The authors should provide a list of what these attributes were to confirm that steps were taken to prevent the dumping effect. Furthermore, in the sample preparation section, the authors should clarify how long the samples were stored before they were analyzed via descriptive test and rheology as the viscosity of starch thickened solutions changes over time.

Reviewer #3 (Remarks to the Author):

The authors propose a model that links the sensory perception of “thickness” in fluids to their shear-thinning rheology. The authors do this by modeling the squeeze flow between the tongue and palate in the oral cavity and comparing it to samples with varying viscosities whose sensory “thickness” and shear-thinning rheology are determined. Unlike previous studies the authors do not assume a single shear rate but account for a dynamic flow. The authors claim significance in that their data support a logarithmic relationship as opposed to a free power law.

I commend the authors for working to bring clarity to an important problem.

My main objection to this paper is as follows:

The authors emphasize that a key point of the paper is the conclusion that there is a logarithmic relationship (ie Weber-Fechner). However, the free power law (ie Stevens), also works, and the evidence that it should be one over the other seems very limited (this is also admitted by the authors on p.5). In particular, it is the two samples with low and high thickness (samples 1 and 8) that deviate the most from the logarithmic fit. More data points at low and high “thickness” — and ideally using other thickeners in addition to various combinations of xanthan gum and “starch” — would make the argument much, much stronger.

Any model on mouthfeel that is based on only two different thickeners (in this case xanthan gum and starch) is bound to have limitations. Thickeners can vary greatly in how they are perceived.

Ideally, at least a couple more thickeners should be included to make the argument stronger. However, this is less imperative than confirming behavior at low/high “thickness”.

In addition, the “methods section” needs to be updated substantially.

Please see below for specific comments.

p. 2, right column, lines 10-12 from bottom: This paragraph is confusing. Please clarify what you did. From the methods I gather you *prepared* some samples and that others were *commercial*. But in this paragraph you start by introducing the test liquids and say they were all commercial. What did you actually do?

p.2 left column, figure legend to Table 1: here you say that fat is fixed at 0.4% but I see no reference to fat in the methods. What “starch” are you using? This table legend would be clearer if it clearly mentioned what Set 1 and Set 2 are. It would also be clearer if you mention that thickness scores were obtained from a sensory panel.

p2. right column, second paragraph: You say here that “the differences between the various liquid bouillons are mainly due to the thickness used”. Yet, here you only test two thickeners in combinations.

p3, Fig 2 and top left two paragraphs: this is great.

p5 top two left paragraphs: given that you emphasize that a key point of the paper is the logarithmic relationship, this does not sound very convincing.

Methods: Methods section need a lot more detail

Sample Preparation:

What is a “classic sauce pan”? specify brand and material

What starch did you use? And why did you choose the one you did?

What brand of starch and xanthan gum?

Specify what you mean by “pure water, incl brand if applicable. Xanthan gum is highly ion dependent so this is important.

How did you select the ratios in table 1?

The figure legend to Table 1 references keeping the fat content at 0.4%. Methods show no mention of fat nor how you ascertained it was constant. How did you ascertain that fat was constant if you don't know the composition. Include this.

What were the store bought commercial samples? What brand?

Descriptive Test:

A better word may be "Sensory Analysis"

You say the panel was "highly trained". How was it trained?

What was the temperature of the samples as they were served? How was it ascertained temperature was kept constant?

On p 4 left paragraph: you say the testing is done with a table spoon of 4 ml — should say that here too.

line 7 from bottom: You say 16-point scale but "0-15" within parenthesis AND your figure 3A shows thickness ranging from 0-10. Clarify what you did.

line 6 from the bottom says: "the panel was trained to score the intensity of all attributes according to salt solution references". What were the references? How were they prepared/concentration? Explain.

Rheology measurement:

You say you performed rheology at 40C, but your sensory panel was served the bouillons at 62C? Explain.

The methods section should also include how you performed the tests for "wetting properties" in figure 2 incl how you determined the angle of 30°.

Typos:

p. 1, left column, line 23: sentence starting "This makes that" sounds incorrect. Please review

p. 1, left column, line 14: should be "liquid textures' " ie possessive, not "liquid textures"

p. 2, fig 1 legend: last line should be "as a function" not "as function"

p. 5, Methods, left column, line 15: delete "that"

p.6, left column, line 17 from top of page: should be "three times" not "three time"

REVIEWER COMMENTS

Reviewer #1 (Remarks to the Author):

Review report on the manuscript entitled “How ‘thick’ are ‘thin’ liquid food products? Predicting thickness perception from their rheology” by A. Deblais et al.

In their manuscript, Deblais and coauthors address the general problem of how humans perceive with their oral cavity differences in the texture of food products. It is a problem that directly falls within current researches that aim at elucidating how human senses function and what sets their sensitivity. More specifically, Deblais and coauthors investigate the relationship between the perceived thickness of liquid bouillons that have a shear-thinning rheological behavior and the elicited shear stresses at the surface of the tongue. This relationship is obtained by combining the results of psychophysics experiments with a human panel and the predictions for the shear stress at the surface of the tongue. The latter are obtained by modeling the liquid flow in the gap between the tongue and the palate with lubrication theory and by using actual rheological measurements of the liquids that are tested. The model, in particular, aims at reproducing the testing conditions, by incorporating both squeezing and shearing of the liquid. Deblais et al. find that the intensity of the perceived thickness increases with the calculated shear stress on the surface of the tongue and claim that a logarithmic dependence better fits their data, in agreement with the well-known Weber-Fechner’s law already derived for other modality senses.

Overall, the paper is well written, the problem is well motivated and of interest. However, I feel that the results that are presented here do not constitute a significant advance in the field. They lack some experimental proof, the theoretical derivations are rather limited and detailed comparisons to previous models are missing, all reasons combined that cannot justify publication of this work in such a high-end journal as Nature Communications. I also feel that the results presented here would be more appropriate for publication in a more specialized journal.

Reply: We thank the referee for this careful and thorough reading of our paper. We appreciate that he/she/they found our study of interest but regret that the referee considers our work not to be of major progress for the field. We believe that our work brings a major improvement on past works by for the first time fully treating the intricate hydrodynamics of the problem; this is also acknowledged by Referee #2, who states *“This paper introduces some new and interesting research to the field, and it brings up valid criticisms of current methods for correlating instrumental measurements with perceived thickness.”* We think that these findings will appeal to a wide audience: people studying food science, rheology, sensory and nutritional sciences, and bio-physiology and for which only a topical journal like *Nature communications* in our view can reach a broad audience and spectrum of researchers who will benefit from our results. Below, we address all specific comments/remarks point by point (in blue), that we hope, will convince the referee. We have modified the manuscript accordingly (blue and wavy underlined) when necessary.

More specifically, establishing a relationship between the perceived thickness and the shear stresses elicited at the base of the tongue is not new. **(1)** Actually, it has already been performed by previous authors, in particular Kokini et al., even though I acknowledge that it was done for other types of food products. Kokini et al. measured an analog power law behavior over almost two decades in the shear stress. I am thus unsure of how the present work is very different from that of Kokini et al.

(2) The difference I note though, is the logarithmic functional dependence that Deblais et al. claim to have found. This could have been an important outcome of the current work, but given the error bars on their measurements, the data is however equally well fitted with a power law and a logarithm. Deblais et al. are well aware of it but use the rather weak argument that a fit with a single parameter is somehow “the best choice” than a fit with two parameters. They thus conclude that a logarithmic dependence is evidenced. But properly testing such logarithmic dependence would definitely require additional experiments to increase the range of the stress values, in particular at low shear stress values where sensitivity is high as well as at higher shear stress values.

Reply: (1) In our manuscript, we fully acknowledge the pioneer contributions of Demartine and Kokini to the field. However, we would like to stress two important points here. First, the experimental works of Kokini *et al.* consider much thicker and elastic fluids for which the earlier derived model of Demartine does not apply. They consider mostly viscoelastic and visco-plastic materials for which the flow profiles and the flow response under a (dynamic) squeezing flow are different from a simple power law fluid. Consequently, the generated stress in their study is rather wrong considering the simple hypothesis invoked to derive the model. Second, the biophysical parameters (*i.e.*, lingual force, tongue speed, characteristic time) in the model were kept adjustable, consequently adding several more degrees of freedom in their fitting routine and in the interpretation of the results.

(2) To address the comment of the referee, we benefit from recent data obtained by collaborators at Wageningen University conducted by Annelies Blok (added as co-author) and prof. Markus Stieger. Importantly, this new set of data (set 3) has been designed to reach larger values of stress σ while keeping the elasticity contribution negligible to work within the framework of our model (see the compositions of these solutions in the updated Table 1 of the manuscript, and elasticity contribution in the updated Sup. Fig. 1). Consequently, in the revised manuscript the stress ranges from 1 to 110 Pa covering 2 orders of magnitude. In opposition to the two first set of data 1 & 2, this new set of data has been evaluated by a non-trained panel. This implies that for a same sample composition (see e.g., sample 1 of set 3 and sample 2 of set 1), the thickness score has been matched with the thickness of the former sets of solutions evaluated by the highly trained panel. As a result, we found here again that the logarithmic dependence on the shear stress (Weber-Fechner law) describes our experimental data better compared to a power law (Stevens), and over more than a decade of stress. We believe that these new data and results strengthen our conclusion, and we thank the referee for suggesting it.

On the theoretical aspects of this work, I am also skeptical about the novelty of the present derivations as they clearly fall within the same theoretical framework that DeMartine and Cussler had developed for the evaluation of liquids between two fingers. Similarly, the authors solve the lubrication theory equation, however with different boundary conditions to take into account the shear motion of the tongue at velocity V . Even if providing a solution to this equation appears rather tricky and calls for some assumptions, I feel that it does not constitute a major improvement. In addition, the expression the authors find for $h(t)$ (Eq. 13) involves the velocity of the tongue V . How does this expression compare to the result of DeMartine and Cussler? In particular, am I wrong to assume that in the limit $V \rightarrow 0$ (squeeze flow only), one should recover their result?

Reply: We believe that our work brings a major improvement on past works by fully and for the first time treating the intricate hydrodynamics of the problem and in particular the combined effects of shear-thinning and dynamics squeezing. As we have already pointed in the main text, the earlier model proposed by Demartine [38], underestimates the generated stress on the tongue since the

aforementioned effects (dynamic squeezing and “thinning effect”) are not take into account. It is also important to stress that the addition of these two effects have a nontrivial effect on the shear stress σ , depending on the sample composition. Below we show a comparison between the two models that we have also added as a new supplementary figure to which we now refer and comment in the main text.

New added Sup. Fig: Generated stress σ given by our dynamic squeezing model (Eq .5, filled circles) compared to the static squeezing model initially derived by Demartine (ref. [39], empty squares). The generated stress is underestimated in the latter case (“shifted” to the left), and the sample’s rheology (i.e., consistency index κ and power law n) have a non-linear effect on σ .

The referee is right; in the limit, of pure viscous liquid, i.e. $n = 1$, the derived equation 5 in our manuscript simplifies to the case of static squeezing. We mention in the main text “Note that if one considers the specific case of a Newtonian liquid ($n= 1$), the result becomes independent of the tongue velocity V , as expected [49, 50]”.

The current model also allows the authors to provide magnitudes of the elicited stresses at the base of the tongue. **(i)** How do these compare with actual biological measurements? How could this model be tested directly and how is it better than previous models? These aspects should be discussed.

(ii) Finally, this model does not take into account the fact that the tongue itself is much softer than the palate. However, as the gap height tends to zero, one could expect elastohydrodynamics effects to come into play. How would these effects change the overall stress response? I feel that these aspects should also be discussed.

Reply: (i) To the best of our knowledge, no biophysical study reported in vivo values of shear stresses on the tongue when the tongue is sheared against the palate, in opposition to a large number of studies reporting in detail the normal pressure on the tongue when it is pressed against the palate (see e.g., [1,2,3] among many others). Nevertheless, a couple of studies that we are

¹Min Yu and Xuemei Gao, Tongue pressure distribution of individual normal occlusions and exploration of related factors, Journal of oral rehabilitation, 46, 2019.

²Lee, J. H., Kim, H. S., Yun, D. H., Chon, J., Han, Y. J., Yoo, S. D., ... & Soh, Y. (2016). The relationship between tongue pressure and oral dysphagia in stroke patients. *Annals of rehabilitation medicine*, 40(4), 620.

³Kieser, J. A., Farland, M. G., Jack, H., Farella, M., Wang, Y., & Rohle, O. (2014). The role of oral soft tissues in swallowing function: what can tongue pressure tell us?. *Australian dental journal*, 59, 155-161.

referring to in our manuscript, explore the role of the shear stress on the deformation of model papillae⁴; these studies are conducted on liquids varying in viscosities in the same range of viscosities of our samples and show that relevant shear stress values for deformation of the papillae are within the same values of stress obtained with our model $\sigma \sim 1 - 100 Pa$. We have added these elements in the discussion part (results) of the main text.

(ii) : The referee raises a very interesting point related to contact mechanics and deformation of the tongue when in close contact with the palate; the tongue is indeed much softer ($E_{tongue} \sim 2.5 - 10 KPa$)⁵ than the palate ($E_{HardPalate} = 30 - 50 MPa$)⁶. Due to the presence of the liquid in between, this field is called Elasto-Hydrodynamic Lubrication (EHL) and one can calculate how the deformability of the tongue would affect our assumption, namely that we can consider the mouth/tongue system as two parallel rigid plates.

The resulting deformation δ can be calculated from⁷:

$$\delta = \left(\frac{9F_N^2}{16R_{tongue}E_{tongue}^2} \right)^{1/3}$$

Here, R_{tongue} is the radius of curvature of the tongue⁵, and the other parameters are the same quantities as defined in the manuscript and above.

Substituting all numbers, we obtain deformation $\delta = 7$ mm and a contact area of radius $R = 2$ cm. As such, the deformation is much smaller than the characteristic size of the system $R_{tongue} = 5$ cm, confirming the validity of our initial assumption that we made of two parallel plates of radii R , and hence, that Elasto-hydrodynamic effect should not play an important role in this problem. We have changed Figure 2 accordingly and now discuss this interesting point in the main text of our manuscript.

Minor comments

1 – The wording ‘thickness’ appears very soon in the main text (starting with the abstract) and it is only properly defined on page 3 of the main manuscript. To ease the reading, this wording should be defined from the start.

⁴J. B. Thomazo, J. C. Pastenes, C. J. Pipe, B. Le Reverend, E. Wandersman, and A. M. Prevost, Probing in-mouth texture perception with a biomimetic tongue, *Journal of the Royal Society Interface* 16, 1 (2019), 1904.02510.

⁵Napadow, Vitaly *et al.*, A biomechanical model of sagittal tongue bending, *J. Biomech. Eng.* 124, 2002

⁶Choi, Joanne Jung Eun *et al.*, Niels Mechanical properties of human oral mucosa tissues are site dependent: A combined biomechanical, histological, and ultrastructural approach.

⁷Johnson, K. L, 1985, *Contact mechanics*, Cambridge University Press.

2 – On page 6, it looks like the kappa constant is misplaced in Eq. 5. My understanding is that it should be outside the n powered term.

3 – In deriving Eq. 8, assumption is made that the parameter α is much smaller than the velocity V . Looking at Fig. 4 of the Supplemental Information, this assumption appears to be valid at long times only and not for $t > 0.2s$ as the authors write in Supplementary Note 5. The expression the authors find for $h(t)$ (Eq. 13) is thus, I understand, only valid at long times and this should be clearly stated in the main text.

Reply: We thank the referee for all these detailed remarks.

1-We have defined the wording thickness in the abstract of the paper according to the definition used by the food science community.

2-Thanks, we have corrected Eq.5.

3-This is now explicitly mentioned in the new version of our manuscript (page 4).

Again, we thank referee #1 for her/his very useful comments and inputs that helped us to improve the quality of our paper.

Reviewer #2 (Remarks to the Author):

This paper proposes a novel method for modelling perceived thickness of thickened liquids by suggesting a logarithmic relationship between perceived thickness and stress on the tongue. This paper introduces some new and interesting research to the field, and it brings up valid criticisms of current methods for correlating instrumental measurements with perceived thickness. This paper would be suitable for publication with minor revisions as suggested below.

Reply: We thank reviewer #2 for his/her positive comments and we appreciate that she/he found our manuscript suitable for publication. We address her/his valuable remarks below and have corrected the main text accordingly (all changes are highlighted in blue).

In the introduction, it would be good for the authors to discuss the effect of other textural attributes on the perception of thickness, such as stickiness and slipperiness. In addition, the authors should include that besides rheology, there is work being done with tribology that is also trying to correlate instrumental measures with sensory textural attributes.

Reply: In the new version of our manuscript, we have broadened the introduction and describe the effect of other textural attributes on the perception of thickness, added relevant references, and we also now discuss the tribology field and its correlation to sensory attributes such as slipperiness.

In the methods, the authors mention that “the panel was trained to score the intensity of all attributes”. **(i)** The authors should provide a list of what these attributes were to confirm that steps were taken to prevent the dumping effect. **(ii)** Furthermore, in the sample preparation section, the authors should clarify how long the samples were stored before they were analyzed via descriptive test and rheology as the viscosity of starch thickened solutions changes over time.

Reply: **(i)** The attribute list has been added to the supplementary information. In total X attributes were used to describe the sensory properties of the foods, so that dumping has been avoided. In

addition to the descriptive sensory evaluation of these X attributes, an open field text box was provided after evaluating each sensory attribute to allow panelists to remark on perceptions that could not be captured in the predefined list of attributes. We have added this missing information in the Method sections and thanks the referee for suggesting it.

(ii) We agree with the referee that the flow behavior of the solutions might change in time as consequence of the temperature dependance of the thickeners used (e.g., swelling of the starch, conformation of the xanthan gum). Nevertheless, as for the evaluation of the thickness attribute by the panel, shear and extensional rheology measurements of the samples were performed immediately after preparation (< 30 min) of the samples. On this time interval, we checked that the rheology remains unchanged for a low- and a high-concentrated sample. This is now shown in a new added supplementary figure (in Sup. Inf.) and mentioned in the revised methods section.

Reviewer #3 (Remarks to the Author):

The authors propose a model that links the sensory perception of “thickness” in fluids to their shear-thinning rheology. The authors do this by modeling the squeeze flow between the tongue and palate in the oral cavity and comparing it to samples with varying viscosities whose sensory “thickness” and shear-thinning rheology are determined. Unlike previous studies the authors do not assume a single shear rate but account for a dynamic flow. The authors claim significance in that their data support a logarithmic relationship as opposed to a free power law.

I commend the authors for working to bring clarity to an important problem.

My main objection to this paper is as follows:

(i) The authors emphasize that a key point of the paper is the conclusion that there is a logarithmic relationship (ie Weber-Fechner). However, the free power law (ie Stevens), also works, and the evidence that it should be one over the other seems very limited (this is also admitted by the authors on p.5). In particular, it is the two samples with low and high thickness (samples 1 and 8) that deviate the most from the logarithmic fit. More data points at low and high “thickness” — and ideally using other thickeners in addition to various combinations of xanthan gum and “starch” — would make the argument much, much stronger.

(ii) Any model on mouthfeel that is based on only two different thickeners (in this case xanthan gum and starch) is bound to have limitations. Thickeners can vary greatly in how they are perceived. Ideally, at least a couple more thickeners should be included to make the argument stronger. However, this is less imperative than confirming behavior at low/high “thickness”.

Reply: We thank reviewer #3 for her/his thorough reading of our manuscript. We took all remarks to heart, and we below address the two main remarks in detail and have changed the main text (highlighted in blue) accordingly.

(i) To address the first remark, (similar to one of the remarks of reviewer #2) we added a new set of data that has been obtained by collaborators at Wageningen University conducted by Annelies Blok (added as co-author) and prof. Markus Stieger. This new set of data (set 3) has been designed to reach larger values of stress σ while keeping the elasticity contribution negligible to work within the framework of our model (see the compositions of these solutions in the updated Table 1 of the manuscript, and elasticity contribution in the updated Sup. Fig. 1). Consequently, in the revised

manuscript the stress ranges from 1 to 110 Pa covering 2 orders of magnitude. We found here again that the logarithmic dependence on the shear stress (Weber-Fechner law) describe better our experimental data compared to a power law, and over more than a decade of stress. We believe that these new data and results strengthen our initial conclusion, and we thank the referee for suggesting it.

(ii) We agree that ideally, one would vary the thickener to assure that it does not affect the perceived thickness, in comparison with a product showing similar flow curve. Nevertheless, the choice of suitable thickeners that exhibit *weak* elastic contribution, in which our hydrodynamics model remains valid, are limited. Our study has been initially designed to take these effects into account and we are at this moment limited by the candidates. We have added a sentence in the discussion part of the main text, stating that, interestingly, further experiments considering samples with similar flow response should be investigated to confirm the behavior observed with Xanthan gum and starch solutions.

In addition, the “methods section” needs to be updated substantially.

Please see below for specific comments.

p. 2, right column, lines 10-12 from bottom: This paragraph is confusing. Please clarify what you did. From the methods I gather you *prepared* some samples and that others were *commercial*. But in this paragraph you start by introducing the test liquids and say they were all commercial. What did you actually do?

Reply: We apologize for the confusion. Set 1 was indeed prepared by mixing samples (model foods) in proportion as indicated in Table I. Set 2 consisted of samples prepared on the basis of commercially available soups (the basis of these samples is very close to the commercial samples (Knorr, Unilever). We made the paragraph clearer in the main text and updated the method section and included the newly added sample set 3.

p.2 left column, figure legend to Table 1: here you say that fat is fixed at 0.4% but I see no reference to fat in the methods. What “starch” are you using? This table legend would be clearer if it clearly mentioned what Set 1 and Set 2 are. It would also be clearer if you mention that thickness scores were obtained from a sensory panel.

Reply: We have updated Table 1 with the suggested corrections.

p2. right column, second paragraph: You say here that “the differences between the various liquid bouillons are mainly due to the thickness used”. Yet, here you only test two thickeners in combinations.

Reply: Indeed, depending on the samples, we used potato or corn starches and xanthan gum as thickeners. We have mentioned these specific thickeners in the sentence.

p3, Fig 2 and top left two paragraphs: this is great.

Reply: Thanks!

p5 top two left paragraphs: given that you emphasize that a key point of the paper is the logarithmic relationship, this does not sound very convincing.

Reply: This remark has been addressed in the reply to the referee's main objection above. We have corrected the main text.

Methods: Methods section need a lot more detail

Sample Preparation:

What is a "classic sauce pan"? specify brand and material

What starch did you use? And why did you choose the one you did?

What brand of starch and xanthan gum?

Specify what you mean by "pure water, incl brand if applicable. Xanthan gum is highly ion dependent so this is important.

How did you select the ratios in table 1?

The figure legend to Table 1 references keeping the fat content at 0.4%. Methods show no mention of fat nor how you ascertained it was constant. How did you ascertain that fat was constant if you don't know the composition. Include this.

What were the store bought commercial samples? What brand?

Reply: In the new version of our manuscript and in the supplementary information, we have fully revised by addressing all these details/remarks (in blue) suggested by the reviewer, and we hope that now it is making the method section much clearer.

Descriptive Test:

A better word may be "Sensory Analysis"

You say the panel was "highly trained". How was it trained?

What was the temperature of the samples as they were served? How was it ascertained temperature was kept constant?

On p 4 left paragraph: you say the testing is done with a table spoon of 4 ml — should say that here too.

line 7 from bottom: You say 16-point scale but "0-15" within parenthesis AND your figure 3A shows thickness ranging from 0-10. Clarify what you did.

line 6 from the bottom says: "the panel was trained to score the intensity of all attributes according to salt solution references". What were the references? How were they prepared/concentration? Explain.

Reply: In the new version of our manuscript and in the supplementary information, we have fully revised by addressing all these details/remarks (in blue) suggested by the reviewer, and we hope that now it is making the method section much clearer.

Rheology measurement:

You say you performed rheology at 40C, but your sensory panel was served the bouillons at 62C? Explain.

The methods section should also include how you performed the tests for “wetting properties” in figure 2 incl how you determined the angle of 30°.

Reply: In the new version of our manuscript and in the supplementary information, we have fully revised by addressing all these details/remarks (in blue) suggested by the reviewer, and we hope that now it is making the method section much clearer.

Typos:

p. 1, left column, line 23: sentence starting “This makes that” sounds incorrect. Please review

p. 1, left column, line 14: should be “liquid textures’ ” ie possessive, not “liquid textures”

p. 2, fig 1 legend: last line should be “as a function” not “as function”

p. 5, Methods, left column, line 15: delete “that”

p.6, left column, line 17 from top of page: should be “three times” not “three time”

Reply: Thanks for pointing these typos. Again, we thank reviewer #3 for his very careful and thorough reading of our manuscript, and the typos have been corrected. We believe that she/he helps us to make the manuscript much clearer and we hope she/he will find it now suitable for publication.

REVIEWERS' COMMENTS

Reviewer #1 (Remarks to the Author):

In their revised manuscript, Deblais et al. have carefully addressed and clarified all my concerns. I was particularly pleased by the efforts the authors made to strengthen the main message of their work, which is related to the Weber-Fechner law. With the new set of data at higher shear stresses they provide, they now clearly demonstrate in a much more convincing way the logarithmic dependence of the perceived thickness with the shear stresses at the base of the tongue. The authors also clarified the important differences between their own theoretical derivations and previous works. In the present form, I now feel that this work is much better suited to be published in Nature Communications and as is, I would finally recommend it, once the two minor points listed below have been taken into account.

Minor comments

1 – On Figure 2D of the manuscript, please add axes x, y and z.

2 – In the methods section, unless I am mistaken, there is a factor 2 missing in Eq. 9. One should read $c_1 \sim [n/(n+1)]^n * (2 * V/\alpha)^n$.

Reviewer #3 (Remarks to the Author):

The reviewers have adequately addressed the concerns and comments.

REVIEWERS' COMMENTS

Reviewer #1 (Remarks to the Author):

In their revised manuscript, Deblais et al. have carefully addressed and clarified all my concerns. I was particularly pleased by the efforts the authors made to strengthen the main message of their work, which is related to the Weber-Fechner law. With the new set of data at higher shear stresses they provide, they now clearly demonstrate in a much more convincing way the logarithmic dependence of the perceived thickness with the shear stresses at the base of the tongue. The authors also clarified the important differences between their own theoretical derivations and previous works. In the present form, I now feel that this work is much better suited to be published in Nature Communications and as is, I would finally recommend it, once the two minor points listed below have been taken into account.

Minor comments

1 – On Figure 2D of the manuscript, please add axes x, y and z.

Reply: We thank again the referee for this careful and thorough reading of our paper; her/his/their comments and inputs helped us to improve the quality of our paper. We appreciate that he/she/they found our study now suitable to be published in Nature Communications. We have changed figure 2 in the new version of our manuscript according to this suggestion.

2 – In the methods section, unless I am mistaken, there is a factor 2 missing in Eq. 9. One should read $c_1 \sim [n/(n+1)]^n * (2 * V/\alpha)^n$.

We carefully checked our derivation of Eq.9 and we do not find a factor 2, and this, if we use coordinates with **(i)** $z \in [0, h]$ and **(ii)** $z \in [0, h]$ and $z \in \left[\frac{-h}{2}, \frac{h}{2}\right]$:

(i)

$$U_x(z) = \frac{hn}{n+1} \sqrt[n]{\frac{h}{m} \frac{\partial P}{\partial r}} \left[\left(c_1 + \frac{z}{h} \right)^{\frac{n+1}{n}} + c_2 \right]$$

With $\alpha \equiv \left(\frac{hn}{n+1} \sqrt[n]{\frac{h}{m} \frac{\partial P}{\partial r}} \right)$ and $k \equiv \frac{n+1}{n}$

$$U_x(z) = \alpha \left[\left(c_1 + \frac{z}{h} \right)^k + c_2 \right]$$

The boundary condition $U_x(z = 0) = 0$ gives:

$$c_2 = -c_1^k$$

which yields

$$U_x(z) = \alpha \left[\left(c_1 + \frac{z}{h} \right)^k - c_1^k \right]$$

The boundary condition $U_x(z = h) = V$ gives:

$$V = \alpha \left[(c_1 + 1)^k - c_1^k \right] \text{ that we solve for } c_1$$

Because $\frac{V}{\alpha} \gg 1$, we have $c_1 \gg 1$. Expanding $(c_1 + 1)^k$ for large c_1 gives:

$$(c_1 + 1)^k \approx c_1 + k c_1^{k-1}$$

That leads to $\frac{V}{\alpha} \approx c_1^k + k c_1^{k-1} - c_1^k \approx k c_1^{k-1}$

Substituting $k \equiv \frac{n+1}{n}$ back:

$$\frac{V}{\alpha} = \frac{n+1}{n} c_1^{\frac{1}{n}}$$

And finally, the result shown in Eq.9 of the manuscript:

$$c_1 = \left(\frac{n}{n+1} \right)^n \frac{V^n}{\alpha^n}$$

(ii) The boundary condition: $U_x \left(z = -\frac{1}{2}h \right) = 0$ gives:

$$c_2 = -\left(c_1 - \frac{1}{2} \right)^k$$

which yields to

$$U_x(z) = \alpha \left[\left(c_1 + \frac{z}{h} \right)^k - \left(c_1 - \frac{1}{2} \right)^k \right]$$

The boundary condition $U_x \left(z = \frac{1}{2}h \right) = V$ gives:

$$V = \alpha \left[\left(c_1 + \frac{1}{2} \right)^k - \left(c_1 - \frac{1}{2} \right)^k \right] \text{ that we solve for } c_1$$

Because $\frac{V}{\alpha} \gg 1$, we have $c_1 + \frac{1}{2} \gg 1$ and $c_1 - \frac{1}{2} \gg 1$, $c_1 \gg 1$

Taylor expansions for large c_1 gives:

$$\left(c_1 + \frac{1}{2} \right)^k \approx c_1^k + \frac{1}{2} k c_1^{k-1} \text{ and } \left(c_1 - \frac{1}{2} \right)^k \approx c_1^k - \frac{1}{2} k c_1^{k-1}$$

Therefore, we find:

$$\frac{V}{\alpha} \approx c_1^k + \frac{1}{2} k c_1^{k-1} - \left(c_1^k - \frac{1}{2} k c_1^{k-1} \right) = k c_1^{k-1}$$

and by resubstituting k , we finally obtain back the same equation Eq.9 as in the methods section:

$$c_1 = \left(\frac{n}{n+1} \right)^n \frac{V^n}{\alpha^n}$$

Reviewer #3 (Remarks to the Author):

The reviewers have adequately addressed the concerns and comments.

Reply: We would like to thank again the reviewer for her/his/their valuable inputs that helped us to make our manuscript clearer.